# Detection of Neuroendocrine Tumours by Enteroscopy: A Case Report

**DOI:** 10.3390/medicina59081469

**Published:** 2023-08-16

**Authors:** Adriana Ortega Larrode, Sergio Farrais Villalba, Claudia Guerrero Muñoz, Leonardo Blas Jhon, Maria Jesus Martin Relloso, Paloma Sanchez-Fayos Calabuig, Daniel Calero Baron, Andres Varela Silva, Juan Carlos Porres Cubero

**Affiliations:** Department of Gastroenterology, Fundación Jiménez Díaz University Hospital, 28040 Madrid, Spain; sfarraisv@quironsalud.es (S.F.V.); claudia.guerrero@quironsalud.es (C.G.M.); lblasjh@fjd.es (L.B.J.); mjmartin@fjd.es (M.J.M.R.); psanchez@quironsalud.es (P.S.-F.C.); dcalerob@fjd.es (D.C.B.); alvarela@quironsalud.es (A.V.S.); jcporres@fjd.es (J.C.P.C.)

**Keywords:** NET, capsule endoscopy, enteroscopy, case report

## Abstract

We present the case of a 62-year-old patient who developed melenas and in whom conventional endoscopic tests could not detect any bleeding lesion. In our case, capsule endoscopy and enteroscopy were the pivotal elements in establishing the diagnosis of a neuroendocrine tumour with an atypical location. As a result, it was possible to surgically remove the lesions at an early stage of the malignancy without metastatic disease and without the need for adjuvant therapy. Our case demonstrates the need for these new techniques in tumours of atypical location and aggressive course. Otherwise, this malignancy may be underdiagnosed until an advanced stage.

## 1. Introduction

Gastrointestinal (GI) neuroendocrine tumours (NET) are rare neoplasms that arise from specialized neuroendocrine cells. Generally, this malignancy can be found anywhere along the gastrointestinal tract and has a poor outcome. They represent a diverse group of neoplasms closely related to the presence of autoimmune atrophic gastritis [1,2]. These tumours represent a therapeutic challenge where endoscopic studies are a fundamental element for their diagnosis [1]. NETs have gained increasing recognition and attention in recent years due to their unique characteristics, complex classification, and diverse clinical presentations. Due to their atypical location, lesions located in the small intestine and colon are usually identified at a late stage, when the disease is at an advanced stage and with widespread dissemination [3]. The development of video capsule endoscopy (VCE) and enteroscopy has facilitated the visualisation of otherwise inaccessible lesions [4]. Our case is a sample of this challenging scenario in which a 62-year-old patient with no previous pathologies developed melenas and in whom previous conventional endoscopic tests did not show any bleeding lesion. Enteroscopy was the decisive and final technique to reach diagnosis.

## 2. Case Presentation

A 62-year-old male with no past medical history was admitted to hospital two months prior to the current episode with obscure gastrointestinal bleeding (OGIB) in the form of melena. The patient had been taking non-steroidal anti-inflammatory drugs for several days without taking proton pump inhibitors. On arrival at the center of origin, the patient had a hemoglobin of 100 g/L with a mean corpuscular volume of 67 fl.

An upper endoscopy was performed to rule out the presence of gastric lesions, showing the presence of bulbitis with no other findings. Therefore, a colonoscopy was performed, in which only sigmoid diverticula were visualised, showing the presence of hematic debris from the small intestine. No bleeding point was found in the colon.

However, having a strong suspicion that a bleeding lesion in the small intestine was causing the melena, a VCE was performed, which showed a subepithelial lesion in the middle jejunum of about 6–7 mm in size with no hematic debris. Upon remission of the bleeding, the patient was discharged with treatment with oral iron and proton pump inhibitors. After outpatient management with oral iron, hemoglobin improved to 115 g/L one month later. In addition, an abdominal computed tomography scan was performed, which failed to identify signs of active intestinal bleeding but showed mild changes related to acute diverticulitis at the descending colon–sigmoid junction. For a more accurate diagnosis, he was referred to our center for further study.

On arrival at our hospital, the patient showed normal hemoglobin levels of 130 g/L with intense iron deficiency with a 2% iron saturation index, elevated transferrin of 413 mg/dL and ferritin of 14 ng/mL. Elevated circulating Chromogranin Alevels of 277.1 ng/mL were found. With the patient’s report and the previous results, it was decided to optimize the diagnosis by performing an enteroscopy.

From proximal to distal ileum (respecting the last cm) around 10 umbilicated, yellowish, submucosal nodular lesions of 3–12 mm, with superficial neovascularization, suggestive of neuroendocrine tumours, were visualized (Figure 1).

Multiple biopsies were taken, and the proximal (oral) edge of the first lesion was marked with activated charcoal (Figure 2).

Enteroscopy allowed to advance along the entire length of the small bowel, even achieving complete visualization of the ileocecal valve from a proximal view (Figure 3).

Pathologist reported the finding of a well-differentiated NET grade 1 with intense and diffuse positivity for chromogranin and synaptophysin with low Ki67 (<3%). Full body Positron Emission Tomography Computerized Tomography (PET-CT) scan, as well as hormone blood tests, were performed for staging. The PET-CT scan was performed using 68Ga-DOTATOC, (a somatostatin receptor-targeted ligand). The study showed lesions along the entire length of the jejunum with several bowel segments involved (Figure 4). After these findings, the case was presented to the Multidisciplinary Committee of Digestive Neoplasms. The surgical management of the lesions was decided together with the General Surgery Department. The patient was referred to them for excision of the affected intestinal segment.

The surgeons performed a supra-infraumbilical midline laparotomy. Upon entry into the cavity and after exploration of the entire abdomen, multiple lesions in the small intestine (ileum) were evidenced. A segment of small bowel was resected (with distal border about 15 cm from the ileocecal valve) including all the suspicious lesions, and a manual latero-lateral anastomosis was performed. The postoperative recovery was satisfactory and the patient was discharged with oral tolerance and digestive transit.

Due to the complete resection of the lesions and the absence of distant disease, the use of adjuvant therapies was not necessary. The patient has not experienced any further episodes of gastrointestinal bleeding with a 5-month follow-up.

## 3. Discussion

NETs exhibit a wide range of clinical presentations, largely dependent on their location, size, functionality, and hormone production. Functional tumours often lead to hormone-related symptoms, known as syndromes. Carcinoid syndrome, characterized by flushing, diarrhea, and wheezing, is commonly associated with GI tract NETs. Insulinoma syndrome arises from pancreatic NETs (pNETs) and manifests as hypoglycemia due to excessive insulin secretion. Similarly, gastrinoma syndrome results from pNETs secreting gastrin, leading to peptic ulcers and acid hypersecretion. Non-functional tumours, on the other hand, may remain asymptomatic until they reach an advanced stage or cause local symptoms due to mass effect. These can include abdominal pain, obstruction, and bleeding, depending on the tumour’s location. The diverse clinical presentations of NETs contribute to the diagnostic challenge they pose [5].

Therefore, the diagnosis of NETs is intricate and necessitates a combination of clinical assessment, imaging studies, and laboratory tests. Imaging techniques play a pivotal role in detecting and localizing NETs. Computed tomography (CT), magnetic resonance imaging (MRI), and somatostatin receptor scintigraphy (SRS) are commonly employed to visualize the tumours and assess their extent [6]. Laboratory tests measure specific biomarkers, such as chromogranin A and 5-hydroxyindoleacetic acid (5-HIAA), which aid in diagnosing and monitoring NETs. Chromogranin A, a protein released by neuroendocrine cells, serves as a valuable marker for disease progression and treatment response. 5-HIAA, a breakdown product of serotonin, is elevated in carcinoid syndrome and assists in diagnosing GI tract NETs. A histopathological analysis of biopsy specimens is essential for confirming the diagnosis and determining the tumour’s grade and stage. Immunohistochemical staining can help identify neuroendocrine markers, further characterizing the tumour [7].

OGIB can be a challenge for clinical practice, causing a need for recurrent medical procedures usually associated with chronic anemization, and serial tests without finding a causal factor [8]. Furthermore, the use of advanced diagnostic techniques such as VCE and enteroscopy has become essential for the management of these conditions, providing the ability to diagnose lesions that were previously inaccessible by other techniques [9].

The small intestine constitutes approximately 90% of the entire gastrointestinal tract. Despite this, tumours located in this region account for only 3% of all malignant gastrointestinal neoplasms. The most frequent malignancies are adenocarcinoma, representing almost 60% of the total, and gastrointestinal stromal tumours, representing about 17%. Sarcomas, lymphomas and NETs are less frequent entities [10,11].

Accordingly, and in line with the above, GI NETs are rare compared to other types of gastrointestinal tumours. However, their incidence has increased in recent decades, possibly due to better detection and diagnosis. It is estimated that GI NETs account for approximately 2% of all gastrointestinal tumours. The average age at diagnosis is around 50 years, and no clear gender predilection is observed [12].

NETs exhibit a wide spectrum of behaviors, histological features, and hormone production capabilities, leading to a complex classification system based on histology and degree of cell proliferation. They are often characterized by their site of origin, such as gastrointestinal (GI) tract NETs, pNETs, lung NETs, and others. Each of these subtypes possesses unique features and behaviors, further emphasizing the complexity of NET classification. The most commonly used classification is the World Health Organization (WHO) classification [13], which divides NET-GI into three main categories: low-grade well-differentiated neuroendocrine tumours (G1), intermediate-grade well-differentiated neuroendocrine tumours (G2) and high-grade poorly differentiated neuroendocrine tumours (G3). This classification helps to predict the biological behavior and prognosis of the tumours. Additionally, NETs are categorized based on their functionality into functional and non-functional tumours. Functional tumours produce excessive amounts of hormones, resulting in distinct clinical syndromes, whereas non-functional tumours do not exhibit hormone-related symptoms [14].

The precise etiology of NETs remains enigmatic, although several factors have been implicated in their development. Genetic predisposition, exposure to certain environmental factors, and sporadic mutations may contribute to the initiation of NETs. In some cases, NETs are associated with inherited syndromes, such as multiple endocrine neoplasia type 1 (MEN1), von Hippel-Lindau (VHL) syndrome, and neurofibromatosis type 1 (NF1), highlighting the role of genetic abnormalities in their pathogenesis [15].

The prognosis of NET-GI varies according to histologic grade, tumour stage, location, and other clinical factors. Low-grade (G1) well-differentiated tumours generally have a more indolent clinical course and a better prognosis compared to high-grade (G3) poorly differentiated tumours. Accurate staging is crucial in determining prognosis and guiding treatment [12,16]. Early diagnosis, accurate staging, and appropriate treatment selection play crucial roles in determining prognosis. Regular surveillance and follow-up are essential for monitoring disease progression, treatment response, and the emergence of any new symptoms. Long-term follow-up is especially important given the potential for recurrence and metastasis, even after successful treatment [17].

Nevertheless, the absence of symptoms until advanced stages means that, in many cases, there is a delay in diagnosis. As a result, liver metastases are seen in 61–91% of cases at diagnosis. In patients with hepatic involvement by malignancy, it is frequent to see, in up to 10% of cases the association with carcinoid syndrome, with varied symptoms such as facial flushing, diarrhea, abdominal cramps, heart valve disease, telangiectasia, wheezing and edema [1,18]. In line with this and since no distant disease was found in our case, our patient did not present any of these symptoms. 

The management of NETs involves a multidisciplinary approach, with treatment strategies tailored to the tumour’s grade, stage, functionality, and patient-specific factors. Surgical resection is the primary treatment for localized NETs, aiming to remove the tumour and prevent its progression. However, for metastatic or inoperable cases, a range of therapeutic options is available [6]. Somatostatin analogs, such as octreotide and lanreotide, are frequently used to control hormonal symptoms and slow tumour growth by binding to somatostatin receptors on the tumour cells. Targeted therapies, including tyrosine kinase inhibitors and mTOR inhibitors, have shown efficacy in certain cases by disrupting pathways that are essential for tumour growth and survival. Peptide receptor radionuclide therapy (PRRT) is an emerging treatment modality that combines a somatostatin analog with a radioactive compound, selectively delivering radiation to neuroendocrine tumour cells. PRRT has demonstrated promising results in managing advanced NETs, particularly those expressing somatostatin receptors. For high-grade, poorly differentiated NETs, chemotherapy and immunotherapy may be considered, although their efficacy is generally limited. Clinical trials exploring novel therapeutic approaches, including immunotherapies and combination therapies, are ongoing and hold potential for improving outcomes [1,19].

VCE is a non-invasive diagnostic procedure that utilizes a small, ingestible capsule equipped with a miniature camera and a light source. As the capsule traverses the gastrointestinal tract, it captures high-resolution images, which are wirelessly transmitted to a data recorder worn by the patient. These images provide a comprehensive view of the small intestine’s mucosa, allowing for the detection of abnormalities, such as tumours, lesions, and bleeding sources. VCE has meant a breakthrough in the evaluation of pathology located in the small intestine. However, this technique does not allow for any type of intervention, facilitating visualization but not biopsy of suspicious lesions. In addition, the dependence of VCE on passive propulsion by peristalsis may result in variable transit times and incomplete visualization [9].

On the other hand, enteroscopy involves the insertion of a flexible, thin tube equipped with a camera and instruments through the mouth or rectum to directly visualize and access the small intestine [4]. Enteroscopy plays a key role in the diagnosis of GI NETs due to its ability to visualize and obtain tissue samples in regions of the small bowel that are difficult to reach with other examination methods. Nevertheless, it requires specialized training and equipment and may not be feasible for all patients [20]. A distinctive point in our case was the ability to reach the entire small intestine, even being able to visualize the ileocecal valve from a proximal view, a differential fact with these new therapies that provides a wide variety of possibilities.

NETs can be located anywhere in the digestive system, but approximately 50% are found in the small intestine, making detection and adequate sampling difficult. Enteroscopy has proven to be a valuable technique for evaluating the small bowel and detecting suspicious lesions [21].

Regarding the diagnostic accuracy of the techniques described above, VCE and enteroscopy would have a diagnostic yield of 10% and 83%, respectively. In cases in which the tumour is hidden in the small intestine, capsule endoscopy may be more sensitive than enteroscopy, although it varies depending on the scope of the observer [1,22].

Due to the higher technical complexity compared to conventional endoscopy, enteroscopy may have a higher risk of complications such as intestinal perforations, pancreatitis and bronchoaspiration. In spite of this, these complications only correspond to 0.72% of all interventions [23].

The utility of enteroscopy in the diagnosis of NET-GI has been supported by some studies. For example, a study by Manguso et al. (2018) evaluated 85 patients with a diagnosis of NET in the small bowel who underwent multiple imaging techniques including CT, MRI, SRS, and enteroscopy. Enteroscopy was the technique that achieved the highest sensitivity with 88% [24].

In addition to screening and diagnosis, enteroscopy also plays an important role in the staging of NET-GI. Accurate staging is essential to determine prognosis and guide appropriate treatment. Enteroscopy can help identify the presence of small bowel metastases and assess the extent of the disease, which is crucial in determining the optimal therapeutic approach [25].

Research in the field of NETs continues to expand our understanding of their molecular mechanisms and potential therapeutic targets. Advances in genomic profiling have provided insights into the genetic alterations driving NET development, enabling the exploration of precision medicine approaches. Immunotherapies, which harness the body’s immune system to target and destroy tumour cells, are a promising avenue for NET treatment. Clinical trials investigating immune checkpoint inhibitors and other immunotherapeutic strategies are shedding light on their potential role in managing NETs. Furthermore, the development of patient-derived xenograft models and three-dimensional cell culture systems is enhancing our ability to study NET biology and test new treatment regimens [26,27].

## 4. Conclusions

In conclusion, enteroscopy plays a pivotal role in the diagnosis of gastrointestinal neuroendocrine tumours. It provides a direct and detailed visualization of lesions in the small bowel, which facilitates the detection and sampling of neuroendocrine tumours. In addition, it contributes to accurate staging of the disease, which aids in treatment planning. Enteroscopy, along with other diagnostic and evaluation methods, is essential to ensure the accurate diagnosis and optimal management of NET-GI, as described in our case. Its ability to provide a complete visualization of the mucosa, contribute to accurate diagnosis and facilitate therapeutic interventions offers substantial advantages to patients. As these techniques continue to evolve, their role in improving patient outcomes and our understanding of these complex tumours will expand further.

## Figures and Tables

**Figure 1 medicina-59-01469-f001:**
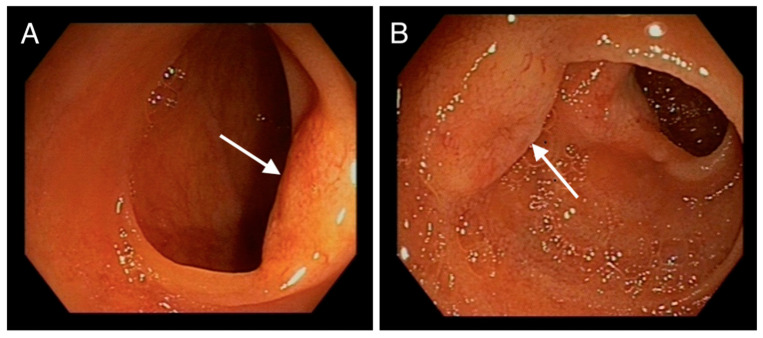
(**A**,**B**): Endoscopic images of jejunum showing nodular, submucosal, umbilicated, yellowish lesions with superficial neovascularization (denoted by arrows).

**Figure 2 medicina-59-01469-f002:**
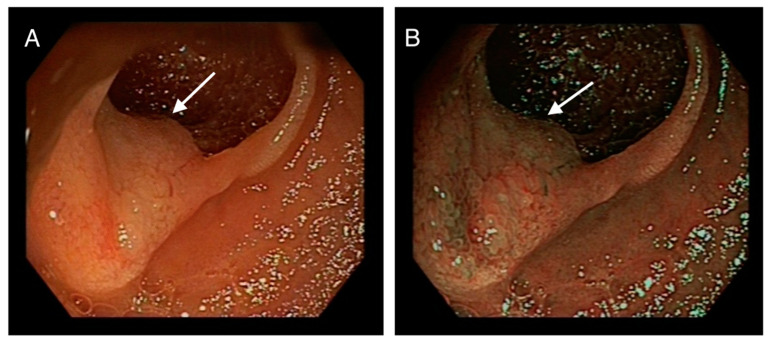
(**A**): arrow pointing at another lesion in the jejunum. (**B**): Same jejunal lesion marked with activated charcoal.

**Figure 3 medicina-59-01469-f003:**
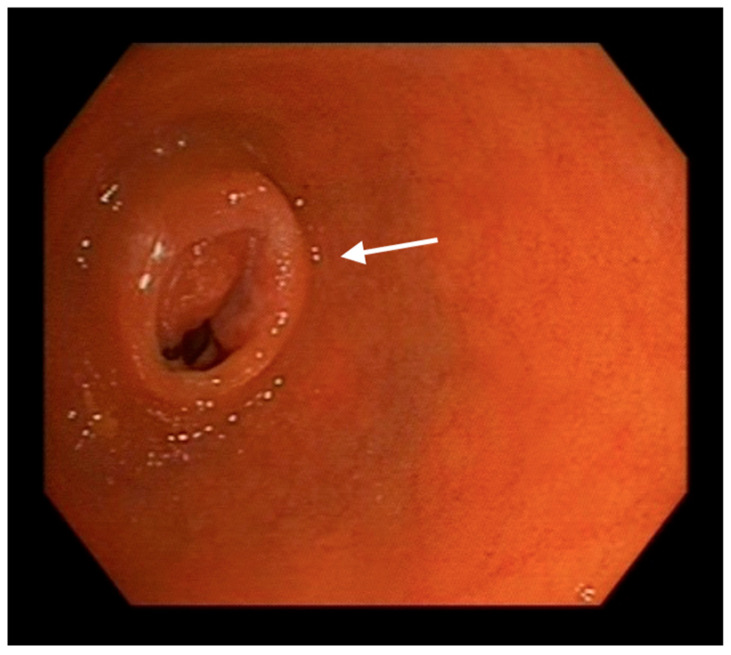
Endoscopic image of the ileocaecal valve (indicated with an arrow) from a proximal (oral) approach reached by enteroscopy.

**Figure 4 medicina-59-01469-f004:**
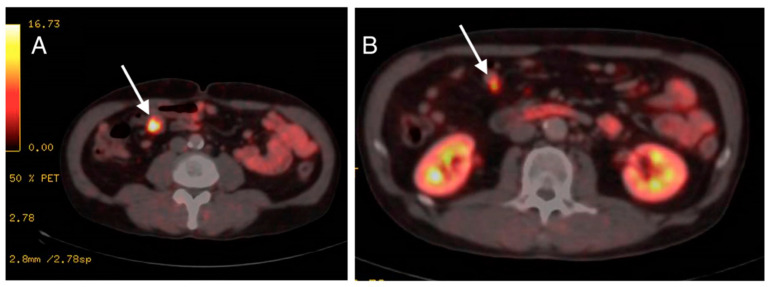
(**A**,**B**): PET-CT scan images showing radiotracer-targeted lesions pointed by arrows.

## Data Availability

Not applicable.

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
