# Peer review of "Detection of Neuroendocrine Tumours by Enteroscopy: A Case Report"

_medicina, 2023, doi:10.3390/medicina59081469_

Round 1

Reviewer 1 Report

The present case report describes a case of 62-year-old male with obscure GI bleeding. UGI endoscopy and Colonoscopy in the case were not suggestive of any lesions. Capsule endoscopy clinched the diagnosis and enteroscopy diagnosed the type and extent of lesion. 

I have many concerns about this paper:

1. The paper is not as per the CARE guidelines of reporting a case. Please follow the guidelines. CARE guidelines for case reports: explanation and elaboration document - PubMed (nih.gov)

2. The study just describes the case till diagnosis. What happened after diagnosis? How was it managed? What is the outcome post-surgery? Please mention about adjuvant treatment/remission/survival?

3. In lines no 110-114, the authors mention about previous studies depicting the role of enteroscopy in NETs. This has to be mentioned in detail.

Overall, the study doesn't add any additional information to the existing literature. There is nothing novel. Moreover, the reporting is not as per the international standards.

Moderate editing is required

Author Response

Response to Reviewer 1 Comments

The present case report describes a case of 62-year-old male with obscure GI bleeding. UGI endoscopy and Colonoscopy in the case were not suggestive of any lesions. Capsule endoscopy clinched the diagnosis and enteroscopy diagnosed the type and extent of lesion. 

I have many concerns about this paper:

Point 1: The paper is not as per the CARE guidelines of reporting a case. Please follow the guidelines. CARE guidelines for case reports: explanation and elaboration document - PubMed (nih.gov)

Response 1:We are sincerely grateful that you have taken the time to review our manuscript. Thank you for these comments and suggestions that make our manuscript gain quality and become more refined with each revision. we have carried out a thorough review of the manuscript. We have again reviewed the CARE standard for case reports. We have made a thorough revision of the article according to these guidelines to meet all the items.

Point 2: The study just describes the case till diagnosis. What happened after diagnosis? How was it managed? What is the outcome post-surgery? Please mention about adjuvant treatment/remission/survival?

Response 2:As you indicate, we had previously focused on the diagnosis without detailing the subsequent therapy. We have rearticulated the case presentation and added the further management of the patient. We have added all the postoperative information and commented on the patient's prognosis and current status.

Point 3: In lines no 110-114, the authors mention about previous studies depicting the role of enteroscopy in NETs. This has to be mentioned in detail.

Response 3:Thanks again for your comments and suggestions. In relation to this point, we have further elaborated on this topic and added relevant information about the role of enteroscopy in relation to NETs.Finally we have revised the English again with a native speaker from the UK to improve its quality.

Reviewer 2 Report

Dear authors,

Thank you for the case report you submitted. Your case presentation is helpful to the readers, but some additions and edits are needed. After completing these, I think your research should be re-evaluated.

Specific comments

Abstract

-In this section, you only briefly mentioned the background, patient characteristics and the case, but in order to attract the attention of the readers, I suggest you also mention your findings and results during the case.

Introduction

-This section is very shallow and short. My suggestion is to give more information about gastrointestinal neuroendocrine tumors in this section.

Although it is a rare situation, the use of endoscopic methods at the stage of diagnosis and whether there are different approaches in the methods used, the margins of error in the diagnosis, etc. In cases where you can add literature supported.

It is important to arouse curiosity about the content of the case by detailing this rare case in order to attract the attention of the readers in the summary and introduction.

I think that the Case presentation and Discussion sections are sufficient.

Best Regards.

Author Response

Response to Reviewer 2 Comments

Thank you for the case report you submitted. Your case presentation is helpful to the readers, but some additions and edits are needed. After completing these, I think your research should be re-evaluated.

Specific comments

Point 1: Abstract-In this section, you only briefly mentioned the background, patient characteristics and the case, but in order to attract the attention of the readers, I suggest you also mention your findings and results during the case.

Response 1: We are sincerely grateful that you have taken the time to review our manuscript. Thank you for these comments and suggestions that make our manuscript gain quality and become more refined with each revision.As mentioned, we had not added relevant information in the abstract that would attract readers.We have reformulated the abstract for this purpose by adding relevant information.

Point 2: Introduction-This section is very shallow and short. My suggestion is to give more information about gastrointestinal neuroendocrine tumors in this section.

Although it is a rare situation, the use of endoscopic methods at the stage of diagnosis and whether there are different approaches in the methods used, the margins of error in the diagnosis, etc. In cases where you can add literature supported.

Response 2: Thank you again for your professionalism and hard work. We have also revised the introduction. We have better structured some parts and added information that we believe makes it more complete.

Point 3: It is important to arouse curiosity about the content of the case by detailing this rare case in order to attract the attention of the readers in the summary and introduction.

Response 3: As you suggest, we have carried out a thorough review of the manuscript. We have made substantial changes throughout the manuscript and emphasized the importance of the case.Finally we have revised the English again with a native speaker from the UK to improve its quality.

Round 2

Reviewer 1 Report

In the revised manuscript, the authors have addressed most of my comments. I would like to congratulate the authors for this work. 

In the results section, the authors have mentioned that "there has been no bleeding since then......."Please write the total follow up duration in this section. 

Minor editing is needed

Reviewer 2 Report

Congratulations. Your case report is ready to publish.